# The Impact of Stroke on the Quality of Life (QOL) of Stroke Survivors in the Southeast (SE) Communities of Nigeria: A Qualitative Study

Gloria Ada Adigwe *, Rachel Tribe, Folashade Alloh * and Patricia Smith

School of Health, Sports and Bioscience, University of East London, Water Lane, London E15 4LZ, UK
* Correspondence: u9603469@uel.ac.uk (G.A.A.); falloh@uel.ac.uk (F.A.)

**Abstract:** Purpose: This study aims to explore the Quality of Life (QOL) amongst ten stroke survivors in the SE communities of Nigeria. Design: Qualitative study using semi-structured interviews. Interpretative phenomenological analysis (IPA) was utilized. Setting: Medical institutes in the southeast communities of Nigeria. Participants: 10 participants ranging in age from 29 to 72 years old. Stroke is typically a life-changing catastrophe, claiming over 55 million lives and disabling 44 million people each year. According to the research, stroke incidence has decreased by 42% in high-income nations worldwide but has increased by 100% in low-income areas such as Africa. Surviving a stroke can be a long-term process that impacts numerous elements of an individual's life. Stroke-related QOL is a major health care issue in Nigeria that has received insufficient attention. The primary objective for survivors is to improve their QOL. Thus, it is critical to understand the true impact of stroke on the QOL of stroke survivors from their perspective. Individual, semi-structured, in-depth interviews with 10 stroke survivors were conducted. An IPA approach shaped the interview process and the analysis of the data. Three main themes arose from the data: an 'unfamiliar self' which illuminated the altered body (unpredictable body), the 'recovery of the embodied self' (the transitional stage of recovery) and the 'reconstruction of the embodied self, which reflected a continuous process of belonging and becoming. The impact of stroke on survivors' QOL was twofold: negative and positive. The physical dimension had the largest detrimental impact on the survivors' QOL, according to the findings. Spirituality, on the other hand, had the most positive impact on survivors' QOL in Nigeria's southeast communities.

**Keywords:** stoke survivors; quality of life; LMIC and HIC (Low Middle Income Country and High Income Country); sub-Saharan Africa; Nigeria



## 1. Introduction

Stroke is the fourth-leading cause of disability-adjusted life years (DALYs) worldwide and the fourth-leading cause of disability among adults; survivors usually suffer from functional disabilities that negatively affects their Quality of Life (QOL) [1,2]. Stroke-related mortality and morbidity continue to be public health problems in many African countries, including Nigeria. Its prevalence in Nigeria continues to increase, with a prevalence of 1.14 per 1000 and a 30-day case fatality rate as high as 40%. Cardiovascular diseases were thought to be diseases of a 'Western' style of life, but more recently, they have become recognised as significant diseases in low socioeconomic status societies [3], as well. However, little is known about the burden and nature of stroke in low-income countries, particularly in sub-Saharan Africa [3,4].

Despite the pressure on public health caused by stroke in Nigeria, there are very few studies of the country's diseases profiles [1,3,5–7]. Studies on the QOL of stroke survivors are equally scarce, particularly in the SE regions of the country [2,4,6]. Approximately 70% of the studies have been performed in the more affluent southwest regions of the country. Ultimately, the assessment of QOL is paramount in health care practice and

research, particularly in Nigeria's poor communities, where evidence-based medicine has become a vital priority of the Nigerian health care system [2,7]. Therefore, studies designed to understand the nature and impact of stroke in the Nigerian population must contribute to managing the disease in the country [1,2,8].

It is believed that this method will help to uncover behaviours and trends in the thought and opinions of the survivors and will offer insight into their cultural trends, perceptions and experiences to gain further awareness and provide a deeper understanding of the impact of a stroke on QOL. It is hoped that the findings will contribute to the body of work and knowledge and develop further understanding of stroke survivors' experiences. In addition, this will aid in promoting QOL, contribute to improving health care infrastructure and providing high-quality rehabilitation programs.

## 2. Methods

The study was conducted during the period September 2020–January 2021. Participants were selected from the main ethnic group in SE communities of Nigeria (the Igbo). Participants were aged 29 to 72 years old (see Table 1 below).

**Table 1.** Participant profiles.

| Study ID | Pseudonym | Age | Gender | Type of Stroke | Year of Stroke | Children |
|----------|-----------|-----|--------|----------------|----------------|----------|
| Participant 1 | Buchi | 72 | Male | Ischemic | 2018 | 6 |
| Participant 2 | Emeka | 54 | Male | Haemorrhage | 2019 | 4 |
| Participant 3 | Amaka | 53 | Female | Haemorrhage | 2012 | 4 |
| Participant 4 | Meka | 60 | Male | Ischemic | 2017 | 5 |
| Participant 5 | Ouchi | 62 | Male | Ischemic | 2018 | 5 |
| Participant 6 | Enosi | 55 | Male | Haemorrhage | 2018 | 4 |
| Participant 7 | Bisi | 29 | Female | Haemorrhage | 2015 | 1 |
| Participant 8 | Joko | 60 | Female | Ischemic | 2015 | 7 |
| Participant 9 | Jopadi | 64 | Female | Ischemic | 2015 | 5 |
| Participant 10 | Tobichi | 58 | male | Haemorrhage | 2018 | 0 |

Design: This interpretative, inductive study is part of a larger research project entitled 'Impact of stroke on the QOL of stroke survivors in SE communities of Nigeria utilising: a mixed method approach'. However, it is critical to investigate survivors' perceptions, attitudes, knowledge, cultural perspectives and beliefs. This will aid in capturing the individual's perceptions regarding QOL. Individual interviews with ten participants were conducted. Data collection methods in IPA studies should be flexible enough to allow topics to emerge; the semi-structured interview has been identified as the method that best provides this flexibility, as stated above. This method will assist in identifying trends in the participants' views and opinions, providing additional insight into the situation [8]. This section of the study employs a hermeneutic phenomenological methodology.

### 2.1. IPA

IPA is a qualitative research approach founded by Smith (1996) and was chosen as the methodology for this aspect of the research study. IPA aims to provide detailed examinations of the lived experiences of the stroke survivors via semi-structured interviews with the survivors. It has a focus on persons-in-context in that experiences reported are reflected on within the wider context in which they occur and embraces the intersubjective relationship between the researcher and the participant [9]. Therefore, IPA involves a 'double hermeneutic' where the researcher is making sense of the participant, who is making sense of their world [10]. Within an IPA analysis, the researcher can make various levels of interpretations, but they are all informed by a position of general psychological interest rather than a specific, pre-existing theoretical position, and the interpretations are grounded within the text rather than imported from elsewhere.

### 2.2. Ethical Considerations

Ethics approval was obtained from the following institutions: University of East London (2019 and 2020), Stroke Action Nigeria, Onitsha Anambra State (2020), Chukwuemeka Odumegwu University Teaching Hospital, Amaku Awka (2019) and Nnamdi Azikiwe University Teaching Hospital, Nnewi Anambra State (2019).

### 2.3. Data Collection Methods

The interview questions were open-ended to capture the individuals' thoughts and perceptions in detail concerning their QOL (see Supplementary Material) [11]. Consent forms explaining the process were collected from the participants, who had the option to stop at any time during the interview. The interviews lasted 30–45 min. Interviews were audio-recorded; these were kept alongside the transcripts in a locked safe. The transcripts were read and compared with the audiotape recordings and the field notes several times to verify accuracy.

### 2.4. The Study Population and Recruitment

Ten participants were recruited from a stroke rehab clinic using purposive sampling. This is a sample technique in which the researcher relies on her own judgement when choosing members of the population to participate in the study. A sample size of ten is recommended in the relevant literature [8,12], as this is the sort of range that is necessary to attach meaning to the phenomena being studied [13].

### 2.5. IPA Analysis

All the semi-structured interviews were fully transcribed and analysed using IPA to explore in detail how the stroke survivors made sense of their experiences and to capture the impact of stroke on QOL. This method also uncovers behaviours, trends in thought and opinions of the participants to give further insight into the problems [14]. It also allows for patterns to emerge from the data to create themes [15].

The analysis involved a systematic protocol that allowed for data to be deconstructed to facilitate the development of themes. This involved four phases:

Phase 1—Familiarisation (Multiple reading and making notes);
Phase 2—Sensemaking (Transforming messages into emergent themes);
Phase 3—Theory building (Seeking relationships and clustering themes);
Phase 4—Data refinement and analysis (Writing up an IPA study).

## 3. Summary Results

The three main themes are shown in Table 2.

**Table 2.** Emergent Themes.

| | Master Themes | Superordinate/Sub-Themes |
|---|---|---|
| 1. | **An Unfamiliar Self** | **Unpredictable Body**<br>Self-identity<br>Disempowerment: Physical and psychological<br>Physical Disability |
| 2. | **Recovering of the Embodied Self** | **Transitional Stage**<br>Rehabilitation /Physical Therapy Treatment<br>Education/Management of Condition<br>Faith/Religion and Cultural Beliefs |
| 3. | **Reconstruction of the Embodied Self** | **My New Life after Stroke**<br>Familiar Self<br>Self-discovery<br>Adjustment and Acceptance |

Source: Author.

The IPA analysis resulted in the development of three main themes and several superordinate themes. As the themes emerged, the essence of stroke's impact on the QOL of stroke survivors became clear.

## 4. Thematic Results and Discussion

### 4.1. Master Theme 1: An Unfamiliar Self

#### 4.1.1. Introduction

The meaning of physical changes following a stroke can be interpreted as living with an altered sense of self or an unfamiliar self [16]. According to studies, stroke survivors view their bodies as fragile, unfamiliar, and unreliable. Most of the participants in the study struggled with not being who they used to be and repeatedly referred to themselves as the person before and after the stroke. They described their bodies as being distant and in conflict with their previous and current self-concepts [17]. Although most of the participants described their self-awareness as remaining constant throughout their recovery following stroke, they all described their bodies as becoming unpredictable, strangely foreign, and difficult. Figure 1 displays the most common of the themes and sub-themes: an unfamiliar self, which refers to the unpredictability of the stroke condition. Its component sub-themes feature the role of 'self-identify' alongside 'disempowerment (physical and psychological) followed by the impact of physical disability.

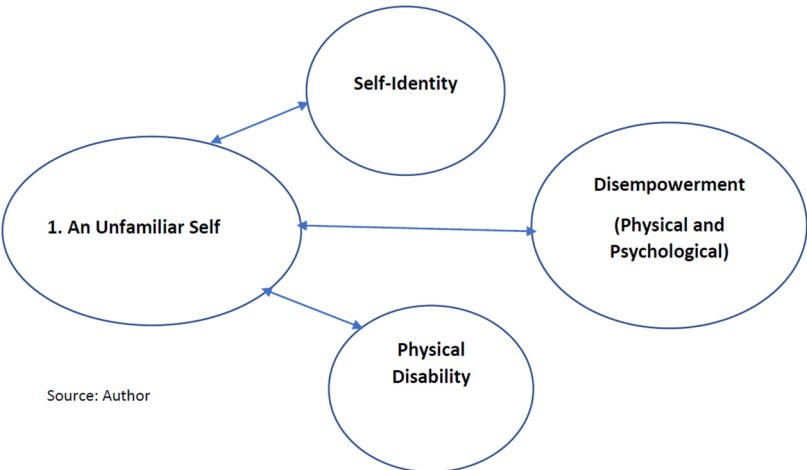

**Figure 1.** An unfamiliar self and sub-themes.

These long-term and often permanent consequences of bodily weakness may turn stroke survivors' focus inwards, away from external activities and projects and relationships with others. Studies that have explored the interrelatedness of bodily changes and perception of self after stroke describe a split between self and the body due to fundamental physical and psychological challenges [18].

#### 4.1.2. Disempowerment (Physically and Psychologically)

Disempowerment, or loss of independence due to various functional disabilities, was the most important factor that had an impact on the QOL of stroke patients [17]. This was one of the major themes discussed by eight out of ten of the participants. This category conveys the participants' loss of physical and functional abilities. The participants often highlighted how they had never particularly thought about how they connected with their bodies before the stroke: 'it was simply normal'. The experience of stroke made the participants increasingly aware of their bodies because of its unfamiliarity [17]. Being unable to move or feel parts of their bodies evoked uncanny feelings of not being the owner of their body parts. The sudden loss of functional ability presents itself in the form of physical limitation resulting from a form of neuronal lesion of the stroke event. Participants

Emeka, Ouchi and Amaka highlighted the sudden physical changes that occurred during the incident:

> *'Then suddenly, noticed my leg and arm become very heavy and my mouth started twisting, that's all I could remember'.* (Emeka, 54 years old)

> *'it becomes so . . . soft, my mouth started to twist . . . just like this . . . .so, I couldn't walk again, even . . . on the chair, I was sitting . . . .i almost fell off'.* (Ouchi, 62 years old)

> *'they took me to the hospital. On my get there, I could not get down from the vehicle again'.* (Amaka, 53 years old)

The above quotes demonstrate how a stroke affects the use of both lower limbs in carrying out an activity that was accomplished before the stroke owing to muscular weakness.

The individual aspects are reflected in Emeka's, Ouchi's and Amaka's struggles to cope with and adapt to the sudden impact of stroke during the process of requiring control of and power over their lives [18]. When they referred to body parts that did not function, the words often used expressed detachment, as if the body part was seen as a strange object from outside. Empowerment is a complex concept and may easily be recognised by its absence: powerlessness, loss of control and subordination [19]. The discussions demonstrated that Buchi's, Tobichi's and Amaka's experiences reflected individual aspects of lack of control and disempowerment [7].

> *'I'm not going anywhere any more since my attack'.* (Bisi, 29 years old)

> *'I was going out before, but since I have not been well, I have just been at home'.* (Amaka, 53 years old)

> *'I am feeling sad that there is no one that I am going to see again, I'm feeling sad like a person'.* (Tobichi, 58 years old)

The participants state that the impact of stroke has contributed to their withdrawal from previous activities in society, leading to isolation. Life for the participants is now one of disability and isolation. A study by [20], reported that disability-related stigma negatively affected the QOL of their study participants. The authors further indicated that the shame attached to having an altered walking pattern led to the social withdrawal of stroke survivors, which inevitably affected their QOL, particularly the younger stroke survivors. This was equally highlighted by the younger stroke survivors in this research study:

> *'or have an idea of what is happening around me'.* (Meka, 58 years old)

> *'it dawned on me that this was serious because there is no other way . . . '.* ( Meka, 58 years old)

> *'and I couldn't leave my bed again, at all (that was all I knew) . . . '.* (Buchi, 72 years old)

> *'I said no, and they agreed and brought a chair, errm stretcher and helped me to the stretcher . . . '.* (Buchi, 72 years old)

> *'I could not drive again . . . so I parked my car . . . '.* (Emeka, 52 years old)

These statements can be interpreted as a lack of control of everyday situations due to the stroke. The above participants reiterate their grief in situations that would previously have been effortless. Meka narrates his loss of awareness during the stroke attack and how scared he felt. Ouchi highlights the severity of his condition as the doctors surrounded him for 24 h a day. The ability of the above participants to talk about their losses demonstrates part acceptance of depending on other people for support. After a stroke, part of the cycle of life is first accepting the loss of control and receiving help from others [21].

The financial burden can cause considerable anxiety and stress to stroke survivors and their families during rehabilitation [22]. The loss of a job or the inability to fulfil previously valued financial roles, therefore immediately failing to provide for the family, often leads to sadness and frustration. The stroke usually means a sudden end to employment. Employment is regarded as one of the most important predictors of QOL. Stroke survivors

who are employed report a better QOL, less health service usage and better health status than non-employed survivors [21,22]. This may result in financial strains, making it more challenging to support the children and family. Emeka narrated the following:

*'when somebody wake up, you cannot have anything to feed their children will lead you to thinking . . . '.*

In Nigerian culture, the man is viewed as the primary provider for the family and is frequently regarded as the family's head. The transition from breadwinner to non-breadwinner is difficult for most individuals and can result in despair. It can be interpreted as a lack of power and independence over their financial condition. In the Nigerian community, having money is seen as important because it earns respect. However, due to the way Nigerian society is structured, people find it difficult to ask friends or relatives for money out of pride.

### 4.1.3. Self-Identification

This section highlights the participants' accounts of issues concerning self-identification, which is frequently linked to a cohesive body-self and an enduring self-identity [23]. The participants indicated that they experienced being socially isolated and were often restricted to their homes. This is a common theme in studies of life after stroke. The body is always present in one's perception of the world, and therefore the self must be embodied [18].This was discovered to have a huge influence on the participants' self–identities. Joko, Enosi, Tobchi, Buchi and Amaka stated the following:

*'there is nothing I am doing, just stay home and do my exercises'.* (Joko, 61 years old)

*'it is because of the stroke I had that is why I am a lot more inactive. I no longer go to places I use to go or enjoy the things I use to enjoy before. I no longer have the freedom I had before . . . erm this has changed my life due to the stroke. I can no longer do the things that I use to do, that I was good at'.* (Enosi, 55 years old)

*'those places I can't attend, have to do without me as I can only do what I can considering my situation'.* (Tobichi, 58 years old)

*'The main limitation is the ability to be active and to go to funerals and burials like I use to before'.* (Buchi, 72 years old)

*'when I get better, I can start going out'.* (Amaka, 53 years old)

As a result of their inability to participate in some activities, participants developed anxiety about their Christian faith. This can also be understood as their bodies becoming untrustworthy, with a break in the link between the body and the self. Most survivors believed that the changes in their bodies threatened their self-esteem and identity; this can be stressful and perplexing. The study of [24], argues that the enforced changes in social positioning involve physical, emotional and social dimensions. Conversely, studies have suggested that the functional limitations of stroke survivors often negatively affect their engagement in daily and social duties [2,14].

Social support has been reported to significantly enhance the QOL of stroke survivors in Nigeria [2,25]. Furthermore, the ready availability of social support in Nigeria is primarily due to the deeply rooted culture of solid kinship ties, communal living and good neighbourliness.

Enosi described himself as an active person before his stroke who was busier but is now inactive. The physical restrictions from his stroke, including muscle weakness, uncoordination and fatigue, caused Enosi to spend more time sitting alone at home. The participants clearly indicated that they were worse off since they had experienced the stroke, and no perception of recovery was expressed. This lack of recovery resulted in participants expressing feelings of uselessness and sadness, where a loss of movement was linked to a lack of recovery.

4.1.4. Physical Disability

Stroke can have an impact on many areas of life, but for most of the participants, physical disability and/or impairment were reported to limit their activities of daily living (ADLs). For the participants, this seemed to have essential and devastating social consequences [17].

This was emphasised by all the study's participants, indicating a negative effect on the participants' QOL.

*'yeah, it has changed me, because before the stroke, I use to cook in my house, but now I cannot without help. Now I need support to cook'.* (Amaka, 53 years old)

Other participants expressed feelings of embarrassment, frustration and difficulty moving. They stated the following:

*'I was going, as I wanted to climb the stairs, but I fell on the floor'.* (Meka, 58 years old)

*'I started limping on the leg, so I went to seek help'.* (Meka, 58 years old)

*'I could not move at all from the bed or use my legs to the private toilet'.* (Emeka, 54 years old)

*'Then I suddenly, noticed my leg and arm become stiff and heavy'.* (Emeka, 54 years old)

*'at times I don't even know or have an idea of what was happening around me, I later tried to move with energy, I couldn't walk. I no longer feel myself it just psychosocial'.* (Buchi, 72 years old)

This can be interpreted as a complete loss of control over one's movement, the incapacity to carry out actions in the same manner as before. The humiliation of becoming powerless, followed by a sense of hopelessness, was rather daunting for the participants. However, other participants saw this as a challenge to be overcome through their efforts; they believed it was their job to recover from the physical consequences of stroke and thus developed personal coping strategies [23].

*4.2. Master Theme 2: Recovering of the Embodied Self (Transitional Stage)*

4.2.1. Introduction

This theme refers to the transition to action in adopting a healthier lifestyle. It embodies the transitional stage of the survivor's experience and is seen as the phase of transitioning from the unfamiliar self to the embodied self-incorporating transition, adaption and healing. Figure 2 displays the recovering of the embodied self, which refers to the transitional stage of the stroke condition. Component sub-themes were rehab/physical therapy treatment, the management of the condition by means of education and the impact of faith and religion.

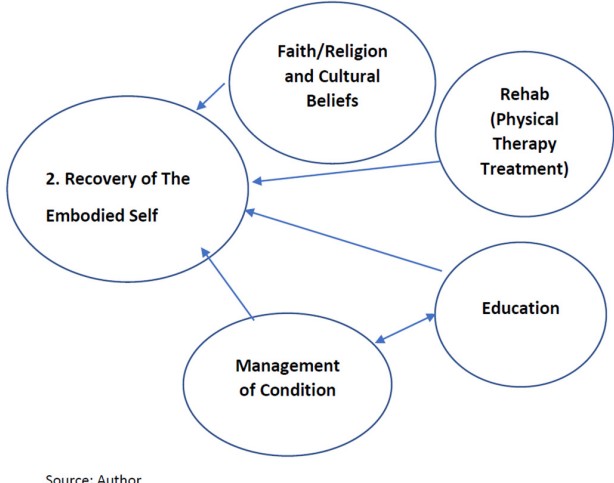

Source: Author

**Figure 2.** The recovery of the embodied self and sub-themes.

This is the stage at which the survivor acquires a better understanding of their condition and begins to show signs of progress through rehabilitation and physical therapy, education and the improved management of their condition, all of which are infused with religion and spirituality. Studies have demonstrated that the reconstruction of the embodied self is related to progress in functional recovery, which is important for QOL [26], whereas other studies have illustrated the interrelationships between body, self and QOL and have described the embodied self as an interwoven relationship of the body and self [17].

The term 'embodiment' is used to refer to how people live in and experience the world through their bodies [17]. Nine out of the ten participants recognised their persistent functional problems and talked about adjusting to the new situation [26]. This consisted of rehabilitation and physical therapy, education and knowledge, better management of the condition and their spirituality.

Stroke survivors attempt to re-establish familiarity with their bodies through a life-long project of testing their limits [27]. The participants in the study experienced a focus on the rehabilitation of physical needs to regain power and control of their lives. Accomplishing significant tasks enabled them to develop a stronger sense of self. Participants reported a need for active participation in their rehabilitation, which was achieved through being aware of their situation and being engaged in independent activities where they could gain a feeling of control over their situation [19]. The participants stated the following:

*'I believe my arm will raise up, because before I was not raising it, because before . . . errh the only thing was the finger, but I am using it little by little. I can now straighten it up to hold something'.* (Emeka, 54 years old)

*'They say it did not touch my bones, it was only soft tissue ie my muscles and nerves'.* (Meka, 58 years old)

*'Yes, I don't exercise for one week, I start to feel bad, exercise is very good for stroke'.* (Amaka, 53 years old)

*'I have started to get up, I can now stand and sit'.* (Tobichi, 58 years old)

*'The exercises have helped me a great deal . . . . A great deal'.* (Enosi, 55 years old)

*'It helped me a bit, but not ready to stand up and walk'.* (Tobichi, 58 years old)

This can be interpreted as participants feeling good about themselves in terms of recovery. However, the best practice in stroke rehabilitation must address all factors that are important to the patient. Studies have mentioned that engagement in rehabilitation helps with recovery [28]. However, rehabilitative interventions conducted one year or more after stroke were reported as not providing any conclusive evidence regarding stroke survivors' ability to return to work. The study [28], found that participants engaged in rehabilitation interventions within eight months after their stroke reported improvement in their health function after they had returned to work.

### 4.2.2. Education

Most of the participants expressed a need for sufficient knowledge and individual information about stroke [18]. Stroke survivors' QOL is negatively impacted by a lack of awareness or education about the disease. They sought information to help themselves during the recovery process and to enhance their QOL. The study by [28] report that gaining education and knowledge helps as a means of controlling feelings of powerlessness. New knowledge helped participants to make sense of why they were experiencing their bodies differently after the stroke, and this understanding often provided reassurance. The participants especially wanted information to help them understand stroke recovery and the rehabilitation process. Participants explained how information helped them to understand rehabilitation goals and to recognise when they were making progress. This provided reassurance that rehabilitation was working [29], whereas the insufficient provision of information prevented them from taking an active part in their rehabilitation. The lack

of information is said to result in a feeling of just 'sitting and waiting for something to happen' [29].

The following participants shared their experiences of unawareness:

*'I know now, I must stop thinking, there is noting that will annoy me anymore, what I know now is that I will manage being upset'.* (Joko, 60 years old)

*'Your diet matters a lot, some people do no eat right'.* (Buchi, 72 years old)

*'I avoid anything that has too much oil in it, we are advised to watch what we eat eg avoid high cholesterol'.* (Emeka, 54 years old)

*'Its HBP and Diabetics, that mainly lead to stroke, but I'm not a diabetic and I don't think I was hypertensive'.* (Bisi, 29 years old)

*'I still have more . . . relief since that day till now, the things I didn't know, now I know'.* (Meka, 58 years old)

These statements suggest that the survivors had learnt a great deal more about stroke disease, particularly the risk factors, and that this knowledge would help in the more effective management of the condition. Education and management of the condition appear to go together, as participants mention how increased knowledge of the condition enabled better and improved condition management.

### 4.2.3. Management of the Condition

Studies report that African Americans, more than others, are at increased risk for hypertension and associated complications [30,31]. Understanding and managing risk factors is crucial with this condition. Observational epidemiological studies suggest that blood pressure is a predictor of both early and late recurrent stroke [10]. It is argued that most patients with ischemic stroke have hypertension; thus, lowering blood pressure may be critical in preventing recurrent stroke [30]. Hypertension is one of the most important risk factors for stroke and is the risk factor with the highest population-attributable risk [10].

The risk factor profile for the participants in the study reflected what is recorded in older individuals in high-income countries with hypertension being the most prevalent risk factor [30]. However, most of the study participants talked about a lack of frequent HBP health examinations and disease control that in some cases brought on the condition.

Out of 10 predominant modifiable risk factors, accounting for 90% of the risk of stroke, hypertension is the strongest [32]. Several studies have demonstrated a very low prevalence of hypertension awareness and control in Africa [31,32].

Participants discussed their lack of previous knowledge and its effect:

*'I suffer from HBP, now I often take my medication. But prior to the stroke I had not taken my medication for a week and some days'.* (Enosi, 55 years old)

*'Everything before now, my BP was very high and the nurses and doctors tried to bring it down'.* (Buchi, 72 years old)

*'Because I am now taking my meds regularly and checking my BP and sugar levels as I do not want my BP to be more than 100 over something. You know they explain all these things to you to better manage the condition'.* (Buchi, 72 years old)

*'I'm advising everyone that are at risk of stroke to go for regular health check-up, stabilise your BP . . . . Go to for you physiotherapy and do not waste time'.* (Buchi, 72 years old)

*'I will be checking my BP often and try to keep my weight down'.* (Jopadi, 64 years old)

*'Now that I have had stroke, I will advise anyone who do not have stroke that they should be careful with their health, they should regularly go for BP check-up if they are at risk as this condition is not a joke. Checking BP regularly so they do not fall into this type of condition'.* (Jopadi, 64 years old)

The above statements are interpreted as poor control of the strongest risk factor, hypertension. Prior to his stroke, Enosi stated that he was inconsistent in taking his HBP medicine

since he felt his condition was not as critical due to diminished HBP symptoms. Buchi also reported that his blood pressure was consistently high and that the medical community fought to bring it under control at one point before his stroke. Emeka emphasised his newfound understanding of the need to maintain his blood pressure and diabetes, and he reported being much more cautious in his maintenance. Jopadi emphasised the necessity of maintaining a healthy weight and getting regular blood pressure readings.

### 4.2.4. Spirituality

Spirituality is centred on values and can be viewed as a source of meaning and purpose for what humans do [16]. This is a key component of a person's daily occupation, and having one's spiritual needs met is, therefore, an essential for engagement and empowerment [28]. This sub-theme had a significant positive impact on the QOL of the survivors. It is an important component and may be a key factor in how survivors cope with the illness and achieve a sense of coherence. According to most of the participants in the study, faith in God was crucial to overcoming the disease.

Most of the participants interviewed had a firm religious belief, which could have been based on their cultural heritage. The study by [33] argues that the lives of healthy people could be enhanced by their spirituality, which gives meaning to their lives and provides comfort in difficult times. The findings of the study [28], similarly express that spirituality helps to support and sustain a positive outlook. Emeka stated the following:

> *'calling God to help us . . . only body that can help us is God. Is only God can do anything that you want in your life . . . '.*

> *'With God anything can be possible just living by the help of other people now, without my junior brother, I don't think how far I would have gone by now because he helped me a lot and he took me to many places to make sure I got better. So, I'm thanking God for that'.*

This is interpreted as the conviction that only God could assist him and that God would eventually heal him. He consoled himself by reminding himself that God is in control of his circumstances. He expressed gratitude to God for the continued support he receives from his family, particularly his brother.

The study by [34] reveals that spirituality can provide vulnerable people who have chronic health conditions with peacefulness, reasons for living, a sense of purpose and a sense of harmony.

Most of the participants were conversant with herbal traditional interventions post-stroke. Traditional medicine (TM) refers to a set of health care practices that are part of that country's own tradition [35]. It is quite common to find people who fall back to TM, especially if their illness concepts are poor.

TM/herbal medicine and its practices play a major role in the health care of the community of Nigeria. In some cases, the native doctor (Babalawo) and the local herbalist (Eleweomo) are the only practitioners available for the treatment of illness. Even in the communities where allopathic medicine is available, the limited facilities it offers make many patients rely on traditional methods [34]. Prior to the introduction of more cosmopolitan medicine, TM used to be the dominant medical system available to millions of people in Africa in both rural and urban communities. Several participants in the study reported using herbal medicine for treatment because they believed it would help them recover faster and because they could not afford their drugs and physiotherapy.

Some of the participants reported that spiritual rituals and or TM improved their health, whereas some of the participants described that TM worsened their conditions.

> *Buchi, Emeka, Joke and Bisi stated the following:*

> *'Yeah . . . herbal medicine. Yeah, they used razor blades to cut my arms and gave me some medicine . . . '.* (Buchi, 72 years old)

*'He gave me some medicine because I was not walking before, then I started walking slowly . . . '.* (Emeka, 54 years old)

*'He used the medicine and he gave me the medicine and I drank it. It get some massage he gave me using some things and hot water to my legs . . . '.* (Joke, 61 years old)

*'Because when I got there, there were people that it was serious issues that are treated and got better, there were people it was very serious for them and they are getting better there . . . '.* (Joke, 61 years old)

*'It helped me sha, in the sense that my hands was in a fist form before, but after taking the TM it stretched out a little . . . '.* (Bisi, 29 years old)

The preceding statements demonstrate that spiritual rituals and TM aided in their health improvement. Buchi recalled how the indigenous doctor sliced open his arms with razor blades and inserted herbal remedies into the wounds. Emeka recounts his initial incapacity to walk, but after receiving the TM, he was able to gently rise to his feet and walk. According to Joke, she was given TM to drink and had her injured leg massaged, which aided her with walking. Bisi reported that her affected hand was in the shape of a fist prior to taking the TM but began to spread out afterward.

*4.3. Master Theme 3: Reconstruction of the Embodied Self*

4.3.1. Introduction

The reconstruction of the embodied self and QOL can be a continuous and intercon­nected process of 'being, doing, belonging, and becoming' [26]. The research indicates that the embodied knowledge of 'how to do things' and 'I can' restored normality and a familiar self without much effort. The narratives in this session led to 'I can', whereas previously it was 'I cannot' and referred to the participants' bodies and abilities to move, perceive, reflect and be aware. This third and final theme indicates reconstruction of the embodied self (as seen in Figure 3). This had a huge impact on the QOL of survivors with sub-themes such as the familiar self, which reflects the 'restoration of the self-body'. Self-discovery reflects 'enhance knowledge and control', whereas adjustment and acceptance reflect 'adaptation and settlement' of the condition.

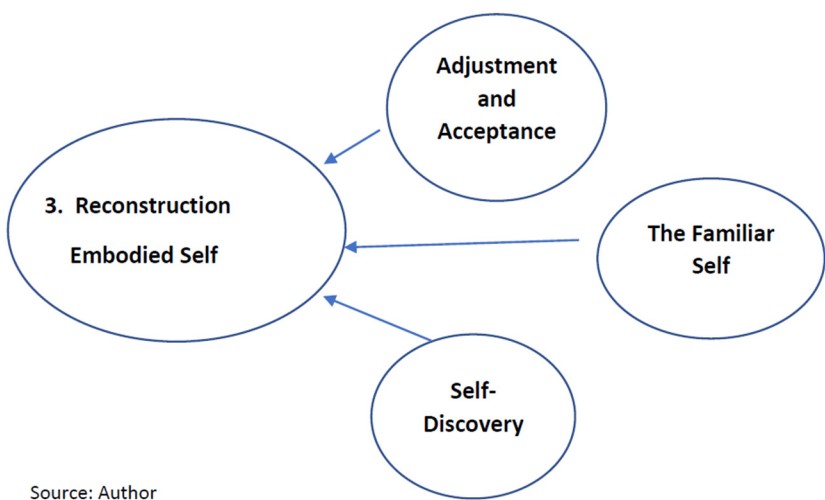

Source: Author

**Figure 3.** Reconstruction of the embodied self and sub-themes.

4.3.2. The Familiar Self

Unlike the strange and unfamiliar sensations and emotions similarly threatened by the union of body and self when one's own bodily utterance cannot be recognised, the familiar self is the restoration of most of the loss of function, reflecting upon a returned normality [17]. This sub-theme was popular amongst the participants of the study and had a positive impact on the QOL of the survivors. The continuous positive change in their

self-awareness, which is driven by a lack of restrictions in functional ability, is a part of recovery [26]. It is assumed, however, according to collected data, that an awareness of the interrelationship between the world, body and self will contribute to a better understanding of how bodily experiences after stroke influences the perception of self [17,36].

The participants who had restored some of their function reflected upon a returned normality, both generally in life and in themselves as individuals. Enosi and Amaka stated the following:

*'But now I walk slightly independently, sometimes with a walking aid but hope soon to walk without any help or aid . . . so to me this is a big achievement'.* (Enosi, 55 years old)

*'It's getting better . . . its becoming better'.* (Enosi, 55 years old)

QOL is central to rehabilitation; however, studies argue that it is a complex, multi-faceted phenomenon [17,19].

### 4.3.3. Self-Discovery

Knowledge of one's own body by discovery in a shared world opens the possibility of understanding other human beings better [17]. Some of the participants in the study struggled in re-establishing the wholeness of their shattered bodies. This sub-theme equally had a positive impact on the QOL of the survivors. Participants' knowledge, as stated earlier, of their post-stroke bodies allowed them to modify everyday activities to allow for changes in their bodily limits, such as the monitoring of fatigue or the use of aids in certain situations, Emeka highlighted the following:

*'After doing a lot of exercises, I have more strength than before . . . errh, this has helped me a lot to get to this level in my recovery'.*

*'When I first came to the rehab club, I saw some people who had worse conditions than myself, I just knew I was going to get better'.*

Self-discovery is also linked with regaining control, which is interpreted as the participants' experience of bodily control and their sense of control over their lives in general. Most of the participants expressed how their stroke had influenced their sense of control until they were able to discover means of bodily wholeness.

### 4.3.4. Acceptance and Adjustment

The reconstruction of the embodied self illustrates a process interconnected with acceptance, progress, adjustment and managing life [17]. This sub-theme had a positive effect on the QOL of the survivors and improved their experiences. Adjustment and acceptance gradually moved forward to a more settled, adapted embodied self [23]. Studies highlight that adjustment and acceptance are interwoven with recovery. Participants in the study attributed their ability to adjust to several factors, such as personal characteristics, diet, lifestyle changes and environment. Bisi and Emeka stated the following:

*'During the sessions I do, I do it with all my ability and strength and I feel I am doing a good job because of all the improvements I have made in my recovery. I feel confident that with time, things will be a lot better for me'.* (Bisi, 29 years old)

*'I'm coping with the assistance of my wife and brother taking care of me'.* (Emeka, 54 years old)

A challenge remains for health professionals to find ways of helping stroke survivors regain a positive perception of self despite changes in bodily functions, social positions, and roles [23]. Bisi reports that:

*'The experiences I have now, is that I can't do those things I used to do easily'.*

The situation of dependency undermines the emotional stability and identity of stroke victims, evoking feelings of futility and failure. The person feels sad, hopeless and less interested in activities that were once enjoyable, slowing the process of recovery of physical and mental health, as stated above by the survivors.

## 5. Conclusions

The overall findings of the study reveal that the physical dimension reflected by an unknown self-had the most significant negative impact on the QOL of survivors, indicating that the participants in the research study feel more deficient in the physical domain, thereby highlighting the need to improve physiotherapy/rehabilitation approaches. On the other hand, the spiritual dimension reflected by reconstruction of the embodied self indicates a beneficial impact on the QOL of stroke survivors in the southeast communities of Nigeria, suggesting that spirituality/faith improves stroke patients' recovery. Consequently, methods and procedures that target the spiritual domain, including 'healing' the spirit, may be advantageous [37].

The interpretative analysis of the impact of stroke on their QOL was marked by situations of loss of independence, reduced self-identity, physical impairment, rehabilitation/physical therapy, education, management of the condition, spirituality, self-discovery, acceptance, and adjustment. Experiencing a stroke suddenly moved people to a state of dependency where they lacked control over their daily lives. Several studies have reported the value of independence and autonomy to participants [37,38].

Most of the participants said that they had, at some point, sensed that something was wrong but thought at the time that it would get better. On the day of the stroke, the perceived symptoms were described as sudden headache and/or loss of control of lower limbs with an inability to move or stiffness or numbness to the upper limb with mouth twisting. Some of the participants sought traditional/native medicine for assistance, while others attended the local hospitals for orthodox medicine. Following hospital treatment, most of the participants returned home with clinical changes such as hemiparesis or hemiplegia and become dependent on others to perform ADLs. Most of the participants expressed fear and concern of becoming dependent on others to perform ADLs, especially the men. These feelings generated a sense of hopelessness among participants due to the disabilities imposed by the stroke attack.

According to several studies [37–40], stroke has an impact on all aspects of health-related QOL (HRQOL), particularly domains in the physical sphere (physical, cognitive, psycho-emotional, and eco-social [ADL] domains). Ultimately, understanding the impact of stroke on the QOL of survivors is paramount in health care practice and research, particularly in Nigeria, where evidence-based medicine has become a vital priority of the Nigerian health care system.

The outcomes of this study will contribute to the advancement of information and knowledge regarding the impact of stroke on the quality of life of stroke survivors in the southeast communities of Nigeria and in poor socioeconomic communities. It is hoped that the findings will enhance awareness of risk factors, particularly among those who are vulnerable. It is believed that the findings will aid in the development of rehabilitation programmes tailored to the activities identified in the study, such as the enhancement of physical programmes, the ability to collaborate with religious institutions and the implementation of strategies that address the spiritual domain in relation to 'healing' and recovery.

### Implications for Practice

One way to evaluate an interpretative analysis's value is to determine the extent to which the insights gained from it contribute to the field [41]. The understanding of what drives stroke survivors' experiences offered by the contextualization of the IPA analysis findings suggests that any implications for practice should occur at a system level. Accordingly, the conclusions of the study should be regarded as our best knowledge of the situation rather than a pronouncement of ultimate reality. The fact that the subjects included young stroke survivors could be a possible restriction. Stroke survivors constitute a heterogeneous group, implying that interviews with other participants may have provided other perspectives regarding the research question. Certain characteristics of QOL would be different for younger persons compared with a more typical-aged group of stroke survivors.

Nonetheless, the findings of this study expand on themes that the clinician working with younger stroke survivors may find useful. Because stroke survivors are a diverse group, interviews with other participants may have provided further insights into the research subject. However, because stroke is impacting younger generations as well, further research is needed to determine the influence of stroke on the QOL of younger stroke survivors.

**Supplementary Materials:** The following supporting information can be downloaded at: https://www.mdpi.com/article/10.3390/disabilities2030036/s1, Open ended questions used for the semi–structured interviews.

**Author Contributions:** Conceptualization, G.A.A.; Data curation, G.A.A.; Formal analysis, G.A.A.; Investigation, G.A.A.; Methodology, G.A.A.; Project administration, G.A.A.; Resources, G.A.A.; Software, G.A.A.; Supervision, R.T.; Validation, G.A.A.; Writing—original draft, G.A.A.; Writing—review & editing, F.A. and P.S. All authors have read and agreed to the published version of the manuscript.

**Funding:** This research received no external funding.

**Institutional Review Board Statement:** The study was conducted in accordance with the Declaration of Helsinki, and approved by the Institutional Review Board University of East London; The Stroke Action Nigeria; Onitsha Anambra State, Chukwuemeka Odumegwu University Teaching Hospital, Amaku Awka; Nnamdi Azikiwe University Teaching Hospital, Nnewi Anambra State (protocol code ETH2122-0015 and 7 September 2021 of approval).

**Informed Consent Statement:** Informed consent was obtained from all subjects involved in the study.

**Data Availability Statement:** The data presented in this study are available on request from the corresponding author. The data are not publicly available due to ethical and privacy reasons.

**Acknowledgments:** The authors would like to thank the medical institutions and rehabilitation centres that were involved in this research study. We are grateful to the staff and the volunteer stroke patients of the Stroke Action Nigeria Onitsha, the Chukwuemeka Odumegwu University Teaching Hospital, and the Nnamdi Azikiwe University Teaching Hospital Nnewi for their active involvement in the study. We would like to thank Christopher Akosile and his internship team for their support during the COVID-19 pandemic.

**Conflicts of Interest:** The authors declare no conflict of interest.

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
