# Peer review of "The Impact of Stroke on the Quality of Life (QOL) of Stroke Survivors in the Southeast (SE) Communities of Nigeria: A Qualitative Study"

_disabilities, doi:10.3390/disabilities2030036_

Round 1
Reviewer 1 Report
It was a pleasure for me to read the manuscript entitled "Impact of stroke on the quality of life (QOL) of stroke survivors in the Southeast (SE) communities of Nigeria: A qualitative study". The manuscript is well-written and appropriately structured. I only have few minor suggestions to improve the quality of data presentation.
1. The authors need to describe in more details the neurological status of the patients enrolled and the time passed after the stroke event. There are different types of stroke and they impact the QoL differently. Also, it might prove to be useful to explain how the authors recruited the patients into the study, cause the "purposive sample technique" is a rather broad approach.
2. The quality of graphic presentation could be further improved. Such, the authors should avoid using dark-blue shading as it is difficult to read the text. Besides, it would be better to use the larger font size and to position the text right in the center of each figure.
3. To make the data collection process more transparent, a separate file with open-ended questions used for semi-structured interviews could be added as a supplementary material.
Author Response
Please see attached cover letter.
Many thanks for your contribution.

Reviewer 2 Report
Overall, the study was well conducted and is appropriate for the journal. The main critique is redundant or unnecessary statements and occasional typos. I have attached the pdf of the article with examples of both highlighted. I recommend removal of the redundant phrases as these hinder reading of the text.

Author Response
Please see attached response to the suggestions made.
Many thanks for your contribution.

Reviewer 3 Report
This study of the impact of stroke on quality of life in Nigeria uses a phenomenological approach to conduct a deep/thick analysis. As such the methods provide important insight into patients experiences and quality of life. However, the article is highly redundant often repeating the same quote in 3 or more different sections. While the division into the three stages of recovery is important, using the same quotes in different sections, undercuts the analytic separation. The spirituality section, in particular is highly redundant. In the results sections, the authors also repeat what is said in the quotes. This method and the “thick” analysis allows quotes to stand on their own so the authors can cut the redundancy.
In regards to organization, Section 3 should be labeled Summary Results. Section 4 should be, “Thematic Results and Discussion.” The authors should either present the total model in section 3 and cut the smaller models in section 4 OR keep the smaller models and cut the larger model in section 3. The conclusion section (5) restates the results in the first section and this should be cut. Because section 4 puts together the discussion and the results, the authors do not summarize how their model is a contribution to prior research. Adding a paragraph with a clear description of how this study goes beyond prior findings would help the conclusions.
Author Response
Please see attached response to your recommendations.
Many thanks.
